# A Privacy-Preserving, Two-Party, Secure Computation Mechanism for Consensus-Based Peer-to-Peer Energy Trading in the Smart Grid

**DOI:** 10.3390/s22229020

**Published:** 2022-11-21

**Authors:** Zhihu Li, Haiqing Xu, Feng Zhai, Bing Zhao, Meng Xu, Zhenwei Guo

**Affiliations:** 1China Electric Power Research Institute, Beijing 100081, China; 2State Grid Corporation of China, Beijing 100031, China; 3School of Electrical and Information Engineering, Tianjin University, Tianjin 300072, China; 4Hangzhou Innovative Institute, Beihang University, Hangzhou 310051, China; 5Key Laboratory of Cryptography of Zhejiang Province, Hangzhou Normal University, Hangzhou 311121, China

**Keywords:** P2P negotiation mechanism, consensus + innovation method, homomorphic encryption, zero-knowledge proof, two-party, secure computation, blockchain, smart contract

## Abstract

Consumers in electricity markets are becoming more proactive because of the rapid development of demand–response management and distributed energy resources, which boost the transformation of peer-to-peer (P2P) energy-trading mechanisms. However, in the P2P negotiation process, it is a challenging task to prevent private information from being attacked by malicious agents. In this paper, we propose a privacy-preserving, two-party, secure computation mechanism for consensus-based P2P energy trading. First, a novel P2P negotiation mechanism for energy trading is proposed based on the consensus + innovation (C + I) method and the power transfer distribution factor (PTDF), and this mechanism can simultaneously maximize social welfare and maintain physical network constraints. In addition, the C + I method only requires a minimum set of information to be exchanged. Then, we analyze the strategy of malicious neighboring agents colluding to attack in order to steal private information. To defend against this attack, we propose a two-party, secure computation mechanism in order to realize safe negotiation between each pair of prosumers based on Paillier homomorphic encryption (HE), a smart contract (SC), and zero-knowledge proof (ZKP). The energy price is updated in a safe way without leaking any private information. Finally, we simulate the functionality of the privacy-preserving mechanism in terms of convergence performance, computational efficiency, scalability, and SC operations.

## 1. Introduction

Recently, renewable distributed energy resources (DERs) [1], electric vehicles (EVs) [2,3,4,5,6], and energy storage systems (ESSs) have turned traditional consumers into prosumers; therefore, they can share energy locally to optimize the load and costs. Many households are now equipped with renewable generators, such as solar panels or wind turbines, which can provide energy in order to satisfy their own demand. The use of these DERs can help more DERs be absorbed into the grid in order to further reduce pollution. However, consumers who participate in the electricity market are required to behave more proactively and are, thus, known as prosumers. The increase in the number of prosumers naturally implies the need for a decentralized energy-trading mechanism that allows prosumers to freely trade with each other without a central supervising entity. Therefore, the network architecture is also changing from centralized to decentralized. A fully decentralized network architecture can be defined as a peer-to-peer (P2P) network in which the participants in the network share a portion of their own resources with one another. These shared resources can be accessed directly by other peers without the intervention of a mediating entity [7]. A formal definition of P2P networks can be found in [8]. In this context, P2P trading mechanisms have emerged as a next-generation energy-management technique that enables prosumers to actively participate in the energy market.

Although the P2P mechanism provides better scalability, reliability, and resilience, growing privacy concerns are hindering its widespread adoption. In a P2P network, it is expected that prosumers will trade their energy with each other without any influence from a central coordinator, which makes P2P platforms a trustless and unreliable system. In addition, P2P energy trading requires a significant amount of data to be exchanged in order to compute the optimal energy amounts and prices for all sellers and buyers [9]. Disclosing such local data for computation would be damaging to their privacy. For instance, local generation reveals the generation capacities and time series of generation patterns [10], and the local demand load reveals consumption patterns [11,12].

Therefore, protecting prosumers’ privacy and encouraging them to cooperate are challenges in such an environment with a lack of trust and security. Different technologies have been used to solve these problems. Blockchain has emerged as a promising, user-friendly, and efficient technology for the implementation of secure and reliable P2P energy-trading mechanisms. Existing studies have exploited a large variety of blockchain-enabled platforms to ensure secure and transparent P2P energy trading [13,14,15,16,17,18,19,20,21,22]. It makes communication transparent for prosumers and allows them to make decisions about energy dispatches in a decentralized and untrusted environment. In blockchain, security mainly means that data are stored on all nodes, are resistant to single points of failure, and are unalterable. Existing blockchain-based energy-trading studies mainly used blockchain to store and protect the final trading results. In addition, smart contracts (SCs) play a very important role in P2P energy trading, as they control the energy transactions between two peers by following predefined rules [17,19,23].

Homomorphic encryption (HE) is a form of encryption that allows for computations on ciphertexts, which generate an encrypted result that, when decrypted, matches the results of the operations as if they had been performed on the plaintext [24]. HE can be further categorized into two classes: semi-HE and fully HE. Semi-HE methods are schemes that only support a subset of the encrypted arithmetic. For example, the Paillier algorithm only supports arithmetic that uses addition; therefore, it is also known as additive semi-HE. On the contrary, fully HE schemes support all encrypted arithmetic. The main advantage of HE is that the security is very high, since it is based on cryptographic techniques, while the most commonly known drawback of HE-based methods is the increased computing power that is required for their complex encryption and decryption operations. Some works have studied the application of HE technology to energy systems. A novel private collaborative distributed energy-management system (P-CoDEMS) was proposed in order to solve the problem of AC optimal power flow (ACOPF) in a distributed and private manner in [25]. Yi et al. integrated HE, blockchain, and other technologies to implement a secure energy trading system [26]. Liu et al. adopted the Paillier method to protect the privacy of ADMM-based distributed DC optimal power flow in [27]. The Paillier-based distributed optimization method was generalized for all gradient-based distributed optimization in [28], and it was reported to be applied to a distributed transactive problem in [29].

However, to our knowledge, existing works did not adequately consider privacy issues in the negotiation process for fully decentralized P2P energy-trading mechanisms. In the P2P energy-trading market, there are multiple agents who negotiate energy trades with each other, and the objective of the market mechanism is to determine the trading prices and amounts for each pair of agents. Thus, in this paper, we propose a privacy-preserving, two-party—instead of multi-party—secure computation mechanism for the negotiation process for each pair of agents. The novel privacy-preserving P2P energy-trading framework combines the technologies of blockchain, SCs, HE, and zero-knowledge proof (ZKP). In detail, we first propose a P2P negotiation mechanism that uses a combination of the consensus + innovation (C + I) method with a power transfer distribution factor (PTDF) model. Then, we analyze the privacy disclosure risk of this mechanism in the case of collusive attacks from neighboring agents. To avoid this risk, a secure, two-party computation framework is designed for updating the energy price between each pair of agents. Finally, the simulation results demonstrate the performance of market convergence, and the line-limit constraints, scalability, and encryption/decryption computation are maintained. The main contributions are the following:We propose a novel P2P negotiation mechanism that incorporates the power transfer distribution factor (PTDF) model into the consensus + innovation (C + I) method, which can simultaneously maximize social welfare and comply with physical line constraints. By introducing line prices into the update process, agents are encouraged not to transfer power over congested lines.Although the C + I method exchanges a minimum amount of information, there is still a risk of revealing private information. We analyze how individual private information (e.g., coefficients of generation, utility functions, and power limits) can be stolen and computed through a collusion attack by a group of collusive neighboring agents in the context of the P2P negotiation mechanism based on the C + I method.The security objective and novelty of this paper are to protect the information exchanged between each pair of agents in the energy-trading negotiation process. We propose a novel, secure, two-party computation mechanism for the energy price update between each pair of agents based on the SC and Paillier encryption algorithm, which is known as an efficient additive HE method. Moreover, we propose a ZKP protocol to prove that the decrypted plaintext matches the ciphertext computed by SC.

The rest of the paper is organized as follows: Section 1 presents the formulation of the P2P energy-trading and social welfare maximization problem. Section 2 proposes the SC-based P2P negotiation mechanism for energy trading, followed by the two-party, secure computation framework in Section 3. The numerical results are presented in Section 4. Finally, in Section 5 and Section 6, the discussions, conclusions, and future perspectives are drawn.

## 2. Problem Formulation

A typical P2P architecture for electricity markets is shown in Figure 1, which consists of simultaneous negotiation of the price and energy of multilateral trades based on predefined trading rules. It can be seen that a P2P mechanism for electricity markets is much more decentralized than existing centralized markets, where all agents must submit all their information, e.g., cost or utility function, power limits, and uncertainty information, to the market operator (MO), who centrally determines the dispatches of energy. In contrast, in P2P markets, all agents can freely negotiate the prices and quantities with each other for multilateral trading.

### 2.1. Peer-to-Peer Trading

In this paper, we build a market with a set Ω of agents defined as either producers or consumers. The market-clearing mechanism proposed below is for a day-ahead market to allocate the supply and demand of energy. It is assumed that all agents are supposed to be rational and truthful, as in [30], which means that they always make decisions to maximize individual benefits. A similar model of the P2P energy-trading process was proposed in our previous work [31,32].

First, the power injection $E_{n}$ of each agent n∈Ω is divided into a sum of bilaterally traded quantities with a set of neighboring agents $m\in \omega_{n}$ as
(1)$$E_{n}=\sum_{\begin{array}{c} m\in \omega_{n} \end{array}}E_{nm},\;\forall n\in \Omega$$

A positive value of $E_{n}$ represents surplus energy and a negative value means required energy. Before P2P energy trading, each prosumer will individually calculate the value of $E_{n}$ according to the power generation and consumption and then decide to be a buyer or seller in the trading. A positive value of $E_{nm}$ represents a sale/production, and a negative value means a purchase/consumption. To lighten notations, $E_{n}$ = $\left\{E_{n1},\ldots ,E_{nm},m\in \omega_{n}\right\}$ is used to represent the whole set of transactions of agent *n*. The power of an agent *n* is constrained as below:(2)$$\underline{E_{n}}\leq E_{n}\leq \bar{E_{n}},\;\forall n\in \Omega$$

Each agent is restrained to either producer or consumer ($\underline{E_{n}}\bar{E_{n}}\geq 0$). Hence, the decision variables are constrained to be positive ($E_{nm}\geq 0$) if it is a producer and negative ($E_{nm}\leq 0$) if it is a consumer, as follows:(3)$$\left\{\begin{array}{cc} \; & E_{nm}\geq 0,\;\forall \left(n,m\right)\in \left(\Omega_{p},\omega_{n}\right)\\ \; & E_{nm}\leq 0,\;\forall \left(n,m\right)\in \left(\Omega_{c},\omega_{n}\right) \end{array}\right.$$
where $\Omega_{p}$ and $\Omega_{c}$ are the sets of energy producers and consumers, respectively.

Finally, the market equilibrium between energy production and consumption is represented by a set of balance constraints of each pair of agents
(4)$$E_{nm}+E_{mn}=0,\;\forall \left(n,m\right)\in \left(\Omega ,\omega_{n}\right)$$

### 2.2. Line Flow Constraints of Power Network

In this paper, PTDF is used to compute the power flow of lines and to label the lines used for power transfer in each transaction [33,34]. The PTDF for line *l* is denoted by $\varphi_{ij}^{l}$ and indicates the fraction of the energy generated by the agents on bus *i* that is transmitted over line *l* to the agents on bus *j*. The PTDF is calculated by $\varphi_{ij}^{l}=\psi_{i}^{l}-\psi_{j}^{l}$, where $\psi_{i}^{l}$, $\psi_{j}^{l}$ are injection shift factors (ISF) in line *l* for bus *i* and *j*. The ISF is an approximation of the sensitivity matrix and quantifies the redistribution of power through each branch after a change in generation or load on a particular bus. The ISF matrix is represented by $\Psi ≜\left[\psi_{i}^{l}\right]\in R^{L\times N}$, where *N* is the number of buses and *L* is the number of lines. This matrix can be obtained using $\Psi ≜B^{\prime }AC^{-1}$ by a diagonal branch susceptance matrix ($B^{\prime }$), a branch-node incidence matrix (*A*), and a reduced nodal susceptance matrix (*C*). In the matrix *A*, $a_{l}^{T}$ is the *lth* row where a line exists between bus *i* and *j*.
(5a)$$\begin{array}{cc} \; & A≜\left[a_{1},a_{2},\ldots ,a_{L}\right]\in R^{L\times N},\;a_{l}^{T}≜\left[0\cdots 0\;\;\overset{i}{1}\;\;0\cdots 0\;\;\overset{j}{-1}\;\;0\cdots 0\right] \end{array}$$
(5b)$$\begin{array}{cc} \; & B^{\prime }≜diag\left[b_{1},b_{2},\ldots ,b_{L}\right]\in R^{L\times L},\;C≜A^{T}B^{\prime }A\in R^{N\times N} \end{array}$$

By having PTDF matrix and traded energy between prosumers, the power flow in line *l* can be computed by (Equation 6)
(6)$$P_{l}=\sum_{n\in \Omega_{p}}\sum_{m\in \Omega_{c}}\varphi_{ij}^{l}E_{nm}$$

In the above Equation (Equation 6), the producer *n* is at bus *i* and the consumer *m* is at bus *j*. Their traded power $E_{nm}$ has an impact on the flow of the line *l*. If the value is below or above the boundaries, the line prices $\left({\bar{\upsilon }}_{l},{\underline{\upsilon }}_{l}\right)$ are sent to the agents using that particular line to transfer power to avoid overflow or congestion.

Since the agents in the power grid use the conventional grid to transmit energy, both social welfare and line flow constraints should be considered. Here, line flow constraints are added as a constraint to the objective function to model the physical network in energy trading. To avoid damage to the transmission lines, the real power flow $P_{l}$ in each line *l* is bounded by the maximum capacity $P_{l}^{max}$ with respect to the heat they can dissipate.
(7)$$-P_{l}^{max}\leq P_{l}\leq P_{l}^{max},\;\forall l\in L.$$

### 2.3. Social Welfare Maximization Problem

To simplify the formulation of the process, we model the production cost and consumer utility functions as quadratic functions of the power set-point, as below:(8)$$C_{n}\left(E_{n}\right)=a_{n}E_{n}^{2}+b_{n}E_{n}+c_{n},$$
where $a_{n}$, $b_{n}$, and $c_{n}$ are predetermined positive constants. From above, the P2P market has the objective to maximize the social welfare of all agents under the constraints. The problem can be equivalently formulated as a cost minimization problem, as below:
(9a)$$\begin{array}{cc} \min & \sum_{\begin{array}{c} n\in \Omega \end{array}}C_{n}\left(E_{n}\right) \end{array}$$
(9b)$$\begin{array}{ccc} s.t.\;\;\; & \underline{E_{n}}\leq E_{n}\leq \bar{E_{n}}\; & \forall n\in \Omega \end{array}$$
(9c)$$\begin{array}{ccc} \; & E_{nm}\geq 0\; & \forall \left(n,m\right)\in \left(\Omega_{p},\omega_{n}\right) \end{array}$$
(9d)$$\begin{array}{ccc} \; & E_{nm}\leq 0\; & \forall \left(n,m\right)\in \left(\Omega_{c},\omega_{n}\right) \end{array}$$
(9e)$$\begin{array}{ccc} \; & E_{nm}+E_{mn}=0\; & \forall \left(n,m\right)\in \left(\Omega ,\omega_{n}\right) \end{array}$$
(9f)$$\begin{array}{ccc} \; & -P_{l}^{max}\leq \sum_{n\in \Omega_{p}}\sum_{m\in \Omega_{c}}\varphi_{ij}^{l}E_{nm}\leq P_{l}^{max}\; & \forall l\in L \end{array}$$

Since the social welfare maximization (or cost minimization) problem is a convex optimization problem, it has a unique optimum that can be achieved by a plethora of centralized methods. However, this requires the disclosure of all the agents’ information. It is better to design a P2P negotiation mechanism that can achieve optimal dispatches of the above optimization problem (9).

## 3. Blockchain-Based P2P Negotiation Mechanism for Energy Trading

In this section, we first design a novel P2P negotiation mechanism for energy trading inspired by the consensus-based approach proposed in [35]. We then present the implementation of P2P energy trading using blockchain and SC.

### 3.1. C + I-Based Decentralized Negotiation Mechanism

The decentralized negotiation mechanism for P2P energy trading is based on the C + I method, which consists of updates to the primary energy quantity variables, updates to the dual variables, and convergence criteria. The main reason for choosing the C + I method to design the market-clearing algorithm is that the information exchanged between agents is minimal compared to other methods, such as the ADMM method [36,37] and the primal-dual gradient [33]. Since the shared information is very small, the communication overhead is lower and the risk of leakage of private information is also lower. Compared with the previous results in [35], the first difference is that the physical line flow constraints of the power grid are considered in our model. Line prices are introduced to induce agents to spontaneously adjust their power generation or consumption, as shown in (Equation 13). The second difference is that SC is used to implement the mechanism, including updating the energy quantities and prices, calculating the power flows, updating the line prices, convergence checking, storing the transaction results, and querying. Therefore, compared with previous work, the mechanism we developed is a more realistic and practical decentralized negotiation algorithm for P2P energy trading.

#### 3.1.1. Local Optimization Problem

For each agent *n* in bus *i*, the local optimization problem at a given iteration *k* is
(10a)$$\begin{array}{cc} \min & C_{n}\left(E_{n}\right)-\sum_{m\in \Omega_{n}}\lambda_{nm}^{k}E_{nm}+\sum_{l\in L}\sum_{n,m\in i,j}^{m\in \omega_{n}}\varphi_{ij}^{l}\left({\bar{\upsilon }}_{l}^{k}-{\underline{\upsilon }}_{l}^{k}\right)E_{nm} \end{array}$$
(10b)$$\begin{array}{ccc} s.t.\;\;\; & \underline{E_{n}}\leq E_{n}\leq \bar{E_{n}} & \end{array}$$
(10c)$$\begin{array}{ccc} \; & E_{nm}\geq 0\; & \forall m\in \omega_{n}\;\mathrm{if}\;n\in \Omega_{p} \end{array}$$
(10d)$$\begin{array}{ccc} \; & E_{nm}\leq 0\; & \forall m\in \omega_{n}\;\mathrm{if}\;n\in \Omega_{c} \end{array}$$
where $\lambda_{nm}$ are the dual variables of the equilibrium conditions (Equation 4) and define the traded energy prices $E_{nm}$. $\lambda_{n}=\left\{\lambda_{n1},...,\lambda_{nm}\right\}$ is used to represent the total traded energy prices between neighboring agents.

#### 3.1.2. Primal Variable Updates

Updates to the energy quantities of agent *n* are based on the Karush–Kuhn–Tucker (KKT) conditions of the local optimization problem. The relaxed Lagrangian function of the local optimization problem (10) at iteration *k* can be expressed as follows:(11)$$\begin{array}{cc} L_{n}^{loc}=\; & C_{n}\left(E_{n}\right)-\sum_{m\in \Omega_{n}}\lambda_{nm}^{k}E_{nm}+\sum_{l\in L}\sum_{n,m\in i,j}^{m\in \omega_{n}}\varphi_{ij}^{l}\left({\bar{\upsilon }}_{l}^{k}-{\underline{\upsilon }}_{l}^{k}\right)E_{nm}\\ \; & +\bar{\mu_{n}}\left(E_{n}-\bar{E_{n}}\right)-\underline{\mu_{n}}\left(E_{n}-\underline{E_{n}}\right) \end{array}$$

According to the first-order optimality conditions of the Lagrangian problem, for all trades between agents n∈Ω and $m\in \omega_{n}$, we have
(12)$$a_{n}E_{n}+b_{n}-\lambda_{nm}^{k}+\sum_{n,m\in i,j}^{l\in L}\varphi_{ij}^{l}\left({\bar{\upsilon }}_{l}^{k}-{\underline{\upsilon }}_{l}^{k}\right)+{\bar{\mu_{n}}}^{k}-{\underline{\mu_{n}}}^{k}=0$$

Then, we can obtain that
(13)$$E_{n}^{k+1}=\frac{\lambda_{nm}^{k}-\sum_{n,m\in i,j}^{l\in L}\varphi_{ij}^{l}\left({\bar{\upsilon }}_{l}^{k}-{\underline{\upsilon }}_{l}^{k}\right)-{\bar{\mu_{n}}}^{k}+{\underline{\mu_{n}}}^{k}-b_{n}}{a_{n}}$$

According to the complementary conditions $\bar{\mu_{n}}\times \bar{E_{n}}=\underline{\mu_{n}}\times \underline{E_{n}}=0$, the above update (Equation 13) can be equivalently transformed to another more concise form, as below:(14)$$E_{n}^{k+1}=\max\left\{\min\left\{\frac{\lambda_{nm}^{k}-\sum_{n,m\in i,j}^{l\in L}\varphi_{ij}^{l}\left({\bar{\upsilon }}_{l}^{k}-{\underline{\upsilon }}_{l}^{k}\right)-b_{n}}{a_{n}},\bar{E_{n}}\right\},\underline{E_{n}}\right\}$$

In this way, the dual variables $\left\{{\bar{\upsilon }}_{l}^{k},{\underline{\upsilon }}_{l}^{k}\right\}$ is omitted and the update process is simpler. Then, the primal variables $\left\{E_{nm},m\in \omega_{n}\right\}$ are updated as below (here for a producer):(15)$$E_{nm}^{k+1}={\left[E_{nm}^{k}+f_{nm}^{k}\left(E_{n}^{(m),k+1}-E_{n}^{(m),k}\right)\right]}^{+}$$
where $f_{nm}$ is an asymptotically proportional factor defined as
(16)$$f_{nm}^{k}=\frac{\left|E_{nm}^{k}\right|+\delta^{k}}{\sum_{l\in \omega_{n}}\left(\left|E_{nl}^{k}\right|+\delta^{k}\right)}$$
with $\delta^{k}$ a positive constant. The operator ${\left[\cdot \right]}^{+}=\max\left(0,\cdot \right)$ in (Equation 15) is used to enforce the sign constraint of the decision variables and is replaced in the case of a consumer by operator ${\left[\cdot \right]}^{-}=\min\left(0,\cdot \right)$.

#### 3.1.3. Dual Variable Updates

The price for a given trade is calculated individually by each agent. After convergence, a consensus has to be reached on these prices (i.e., $\lambda_{nm}=\lambda_{mn}$). The energy price $\lambda_{nm}^{k+1}$ will be updated in this form:(17)$$\lambda_{nm}^{k+1}=\lambda_{nm}^{k}-\beta^{k}\left(\lambda_{nm}^{k}-\lambda_{mn}^{k}\right)-\alpha^{k}\left(E_{nm}^{k}+E_{mn}^{k}\right).$$

Price convergence is ensured in the price update by a consensus term. The last term, the innovation term, ensures energy equilibrium between agents. $\alpha^{k}$ and $\beta^{k}$ are sequences of positive factors set by the individuals such that each excitation is persistent so that the series of each sequence converge. The tuning of these parameters ($\alpha^{k}$ and $\beta^{k}$) is key to the convergence performance of the algorithm and usually requires a trade-off between convergence speed and adaptation to changes in setting. Performance could be improved by using an adaptive parameter. The calculations steps (Equation 13)–(Equation 16) are all performed locally without communicating with others. Only in step (Equation 17) does agent *n* need to receive information $\left\{E_{mn}^{k},\lambda_{mn}^{k}\right\}$ from agent *m* to update the energy price $\lambda_{nm}^{k+1}$.

Finally, the line manager (LM) will be responsible for calculating the power flows in each line by (Equation 6), and the line prices $\left({\bar{\upsilon }}_{l}^{k+1},{\underline{\upsilon }}_{l}^{k+1}\right)$ will be updated as
(18a)$$\begin{array}{cc} \; & {\bar{\upsilon }}_{l}^{k+1}={\left[{\bar{\upsilon }}_{l}^{k}+\phi^{k}\left(P_{l}^{k+1}-P_{l}^{max}\right)\right]}^{+} \end{array}$$
(18b)$$\begin{array}{cc} \; & {\underline{\upsilon }}_{l}^{k+1}={\left[{\underline{\upsilon }}_{l}^{k}-\phi^{k}\left(P_{l}^{k+1}+P_{l}^{max}\right)\right]}^{+} \end{array}$$
where $\phi^{k}$ is the tuning parameter.

#### 3.1.4. Condition of Convergence

The above decentralized algorithm converges as long as the following conditions are met:
(19a)$$\begin{array}{cc} \; & \sum_{n\in \Omega }\sum_{m\in \omega_{n}}\left|E_{nm}^{k+1}-E_{nm}^{k}\right|\leq \chi^{E} \end{array}$$
(19b)$$\begin{array}{cc} \; & \sum_{n\in \Omega }\sum_{m\in \omega_{n}}\left|\lambda_{nm}^{k+1}-\lambda_{nm}^{k}\right|\leq \chi^{\lambda } \end{array}$$
(19c)$$\begin{array}{cc} \; & \sum_{l\in L}\left|{\bar{\upsilon }}_{l}^{k+1}-{\bar{\upsilon }}_{l}^{k}\right|+\left|{\underline{\upsilon }}_{l}^{k+1}-{\underline{\upsilon }}_{l}^{k}\right|\leq \chi^{\upsilon } \end{array}$$
where $\chi^{E}$, $\chi^{\lambda }$ and $\chi^{\upsilon }$ are stopping criterion predetermined by market operator.

### 3.2. Implementation of P2P Energy Trading by Smart Contracts

An illustration of the blockchain-based P2P trading architecture is shown in Figure 2. The process is described below.
In the first step, all agents initiate a pair of energy prices and quantities in parallel and send it to neighboring agents. Then, each agent updates its quantities and prices for its neighbors using (Equation 15) and (Equation 17), respectively. The update process is automatically performed by SC, which is installed on each agent.After updating each agent, all agents send their traded energy to LM, which calculates the power flows and line prices on each line using (Equation 6) and (18), also from SC.Then, LM sends the line flow prices to the corresponding agents using the particular line for power transmission. By applying these line usage price signals, the agents will try to trade energy with nearby ones, which can reduce power losses.After each iteration, each agent and LM send the updated results to MO, who will check if the stopping criteria are met (19).Finally, after the market converges, MO collects all transactions and stores them in the blockchain.

## 4. Privacy-Friendly P2P Computation Framework

We have formulated a decentralized negotiation algorithm between agents based on the C + I method, but there are still obvious shortcomings. During the negotiation process, agents need to share the updated energy and price data with neighboring agents, and privacy may be lost during the process. Malicious attackers can obtain private information by studying the updated energy and prices. Therefore, developing a privacy-friendly information exchange scheme is the prerequisite for P2P energy trading. In this paper, we propose a privacy-friendly, two-party, secure computation scheme, mainly using HE technology, SC, and ZKP to realize secure information exchange between agents. To our knowledge, none of the existing work uses HE for P2P energy trading. Previous works mainly use HE to solve the AC optimal power flow (ACOPF) problem [25], DC optimal power flow [27], and gradient-based distributed optimization [28]. Our work is the first attempt to combine the HE method with a consensus-based approach and to apply it to the P2P energy-trading mechanism. In the proposed scheme, encryption is implemented by the Paillier cryptosystem [38].

There are two security goals for the privacy-friendly P2P computational framework. The first is to protect individual private information $F_{nm}^{k}=\left\{E_{nm}^{k},\lambda_{nm}^{k}\right\}$ from attacks and acquisition by malicious neighboring agents. The second task is to guarantee that the third party (not the agents) follows the energy price update rules (Equation 17) during operation.

### 4.1. Collusion Attack

To perform C + I updates, a minimum amount of information must be exchanged. At each iteration of the process, the set $F_{nm}^{k}$ of information sent from one agent n∈Ω to a neighboring agent $m\in \omega_{n}$ at iteration *k* must be the following:(20)$$F_{nm}^{k}=\left\{E_{nm}^{k},\lambda_{nm}^{k}\right\}$$The internal production/consumption parameters $\left(a_{n},b_{n},\bar{E_{n}},\underline{E_{n}}\right)$ of all agents need not be shared to achieve optimality.

However, this mechanism cannot protect individual privacy. Consider a specific scenario in which the neighboring agents of agent *n* conspire to obtain the internal production/consumption parameters of agent *n*, as shown in Figure 3a. We will introduce two attack strategies to derive the parameters $\left(\bar{E_{n}},\underline{E_{n}}\right)$ and $\left(a_{n},b_{n}\right)$, respectively.
1.If agent *n* is a producer, all neighboring agents (consumers) can intentionally increase the purchase price $\lambda_{mn}$ little by little until $E_{nm}$ remains unchanged between two iterations. In this case, the output of agent *n* has reached the upper bound $\bar{E_{n}}$. After that, all neighboring agents can communicate with each other to sum all $E_{nm}$ and obtain the private information $\bar{E_{n}}$. Similarly, a group of malicious neighboring agents can cooperatively lower the purchase price to obtain the lower bound $\underline{E_{n}}$.2.Since the neighboring agents of agent *n* have received the information about the power boundaries, the group of neighbors for the power update (Equation 13) can construct a set $\lambda_{n}$ such that the output does not reach $\left(\bar{E_{n}},\underline{E_{n}}\right)$, (means $\bar{\mu_{n}}=\underline{\mu_{n}}=0$ ). Under this construction, the update (Equation 13) can be simplified as follows:
(21)$$E_{n}^{k+1}=\frac{\lambda_{nm}^{k}-\sum_{n,m\in i,j}^{l\in L}\varphi_{ij}^{l}\left({\bar{\upsilon }}_{l}^{k}-{\underline{\upsilon }}_{l}^{k}\right)-b_{n}}{a_{n}},\;\forall m\in \omega_{n}$$By substituting two iteration results $\left\{\lambda_{n}^{k},E_{n}^{k+1}\right\},\left\{\lambda_{n}^{k+1},E_{n}^{k+2}\right\}$ (where $E_{n}$ can be obtained by summing up all $E_{nm}$) into (Equation 21), $a_{n}$ can be solved by randomly choosing a trade with neighbor *m*, as below:
(22)$$a_{n}=\frac{\left(\lambda_{nm}^{k+1}-\lambda_{nm}^{k}\right)-\left(V_{nm}^{k+1}-V_{nm}^{k}\right)}{\left(E_{n}^{k+1}-E_{n}^{k}\right)}$$
where $V_{nm}^{k}=\sum_{n,m\in i,j}^{l\in L}\varphi_{ij}^{l}\left({\bar{\upsilon }}_{l}^{k}-{\underline{\upsilon }}_{l}^{k}\right)$, and this is all public information. After obtaining $a_{n}$, $b_{n}$ can be readily calculated by (Equation 21).

Thus, although very little information needs to be shared in C + I updates, there is still the risk of loss of privacy in the event of a clandestine attack by a group of malicious neighboring agents. There is a need to develop a privacy-protection mechanism for P2P negotiations between agents.

### 4.2. Homomorphic Encryption/Decryption Mechanism

The Paillier algorithm implementation scheme is detailed below [39].

*Key generation:* Two prime numbers *p* and *q* are randomly chosen to satisfy gcd(pq,(p−1)(q−1))=1, where gcd stands for the greatest common divisor. Then, N=p∗q and λ=lcm(p−1,q−1) are founded, where lcm stands for the least common multiple. We randomly pick $g\in Z_{N^{2}}^{*}$ to satisfy $gcd(L\left(g^{\lambda }\;mod\;N^{2}\right),N)=1$ and ensure there exists
(23)$$\mu ={\left(L\left(g^{\lambda }\;mod\;N^{2}\right)\right)}^{-1}mod\;N$$
where $L\left(x\right)=\frac{x-1}{N}$. The public key is found as N,g, and the private key is found as λ,μ.

*Encryption Function (Enc):* Let the plaintext message be $m\in Z_{N}$ and the public key be pk; then, the encrypting function is
(24)$$Enc\left(m,pk\right)=g^{m}\cdot r^{N}\;mod\;N^{2}$$
where *r* is a random pad $r\in Z_{N^{2}}^{*}$.

*Decryption Function (Enc):* Let the ciphertext be *c* and the secret key be sk, the plaintext can be computed as follows:(25)$$m=Dec\left(c,sk\right)=\frac{L(c^{\lambda }\;mod\;N^{2})}{L(g^{\lambda }\;mod\;N^{2})}\;mod\;N=L\left(c^{\lambda }\;mod\;N^{2}\right)*\mu \;mod\;N.$$

**Property 1.** (Additive Homomorphic):
*The additive homomorphic property allows the user to operate the message in its ciphertext directly. Assume the two plaintexts are $m_{1},m_{2}$ and the key pair is ${sk}_{i},{pk}_{i}$; then, we have*

(26)
$$\begin{array}{cc} \; & c_{1}=Enc\left(m_{1},{pk}_{i}\right)\equiv g^{m_{1}}\cdot r_{1}^{N}\;mod\;N^{2}\\ \; & c_{2}=Enc\left(m_{2},{pk}_{i}\right)\equiv g^{m_{2}}\cdot r_{2}^{N}\;mod\;N^{2} \end{array}$$

*Obviously, we have $c_{1}*c_{2}\equiv g^{m_{1}+m_{2}}\cdot {\left(r_{1}\cdot r_{2}\right)}^{N}\;mod\;N^{2}$; thus, we can conclude that*

(27)
$$m_{1}+m_{2}\;mod\;N=Dec\left(Enc\left(m_{1},{pk}_{i}\right)⊕Enc\left(m_{2},{pk}_{i}\right),{sk}_{i}\right)=Dec\left(c_{1}*c_{2},{sk}_{i}\right).$$



**Property 2.** (Non-Deterministic):
*The non-deterministic means that a given plaintext can be encrypted into a very large set of possible ciphertexts. This property prevents an adversary from associating ciphertext with observed information.*


### 4.3. Two-Party, Secure Computation

A privacy-preserving, two-party, secure computation framework is designed using HE, ZKP, and SC, as shown in Figure 3b. Before submitting the transaction data to SC, the agents use the public keys generated by the Paillier encryption algorithm to encrypt the aggregated transaction data. The data are in the form of ciphertext, which does not reveal any private information of the agents even if an attacker obtains it. The result of the ciphertext operation matches the result of the plaintext operation Compared to standard public key encryption, it is the simpler method with the same result, but there is no guarantee that agent *n* follows the rules to compute $\lambda_{nm}^{k+1}$. Agent *n* can increase $\lambda_{nm}^{k+1}$ to make more profit but runs the risk of not offering enough goods in real time. The combination of HE and SC costs more computational resources but can guarantee the update of energy prices, fend off privacy attacks, and restore the computation result to the blockchain for verification.

Looking at the update steps, only the energy price update (Equation 17) will use the information $F_{nm}^{k}=\left\{E_{nm}^{k},\lambda_{nm}^{k}\right\}$ received from neighbor *m*. Thus, the energy price update is implemented by the Paillier encryption algorithm since it satisfies additive homomorphic. The HE- based secure two-party computation algorithm is described below.
Agent *n* generates an individual public key $pk_{n}$ and a secret key $sk_{n}$. The public key is sent to agent *m* for encryption.Agent *n* performs an aggregation operation $I_{nm}=\left(1-\beta^{k}\right)\lambda_{nm}^{k}-\alpha^{k}E_{nm}^{k}$, and an encryption $Enc(I_{nm},pk_{n})$ is sent to SC on Agent *n*.Agent *m* also first performs an aggregation operation $I_{mn}=\beta^{k}\lambda_{mn}^{k}-\alpha^{k}E_{mn}^{k}$ and an encryption $Enc(I_{mn},pk_{n})$ using agent *n*’s public key and sends it to SC.After collecting the information from two agents, SC computes $Enc\left(I_{nm},pk_{n}\right)⊕Enc\left(I_{mn},pk_{n}\right)$. From (Equation 17), we have $\lambda_{nm}^{k+1}=I_{nm}+I_{mn}$. Thus, according to the additive homomorphic encryption property, the result is $Enc(\lambda_{nm}^{k+1},pk_{n})$, which will be sent to agent *n* and *m*.Agent *n* executes $Dec\left(Enc\left(\lambda_{nm}^{k+1},pk_{n}\right),sk_{n}\right)$ to obtain the decryption $\lambda_{nm}^{k+1}$ and sends it to Agent *m*.Agent *n* generates and sends a ZKP to Agent *m* to prove that the plaintext $\lambda_{nm}^{k+1}$ is correct with the ciphertext $Enc(\lambda_{nm}^{k+1},pk_{n})$ computed by SC. Details of the construction of the ZKP are provided in Appendix A.

**Remark** **1.**
*Another challenge is to verify the authenticity of the message $Enc(I_{nm},pk_{n})$. To solve this problem, we can take advantage of digital signatures. Agent n first uses a one-way hash function to obtain a 128-bit digest $H(Enc\left(I_{nm},pk_{n}\right))$ and then encrypts the digest with its private key to obtain the encrypted digest $D_{n}=Enc\left(H\left(Enc\left(I_{nm},pk_{n}\right)\right),sk_{n}\right))$. The message $Enc(I_{nm},pk_{n})$, the encrypted digest $D_{n}$, and the public key $pk_{n}$ are packed and sent to SC. SC verifies the authenticity of the message by checking that the digest of the message processed by the hash function matches the decryption of the received encrypted digest with the public key, i.e.,*

(28)
$$H\left(Enc\left(I_{nm},pk_{n}\right)\right)\overset{?}{=}Dec\left(D_{n},pk_{n}\right)$$



### 4.4. Security and Privacy Analysis

Given the two security goals, to achieve the first goal, we first perform an information aggregation operation for agent *n* and *m*, respectively ($I_{nm}=\left(1-\beta^{k}\right)\lambda_{nm}^{k}-\alpha^{k}E_{nm}^{k}$ and $I_{mn}=\beta^{k}\lambda_{mn}^{k}-\alpha^{k}E_{mn}^{k}$). By using aggregation operations, even if attackers obtain the information, they cannot reveal the original information. Then, agent *n* uses public key $pk_{n}$ to encrypt $I_{nm}$ and sends $pk_{n}$ to neighboring agent *m* to encrypt $I_{mn}$. The information is encrypted with the public key of agent *n*, so even if the information is obtained by malicious attackers, the original data cannot be recovered without the private key. The information is encrypted with agent n’s public key, so it is undeniable that agent *n* can recover $I_{m}n$. However, agent *n* can only obtain the value of $I_{m}n$; there is no way for agent *n* to recover the original private information $\left\{E_{mn}^{k},\lambda_{mn}^{k}\right\}$ from $I_{m}n$ since the aggregation operation is performed locally in agent *m*.

To achieve the second goal, the third party is traditionally required to provide zero-knowledge proof of the additional operation. However, this can lead to a higher computational cost for generating the proof. In this work, HE ensures that the decryption value of the result of the ciphertext computation is equal to the result of the plaintext computation, and we use secure SC to realize the ciphertext computation $Enc\left(I_{nm},pk_{n}\right)⊕Enc\left(I_{mn},pk_{n}\right)$. Thus, the combination of SC and HE can ensure the correctness of the result $Enc(\lambda_{nm}^{k+1},pk_{n})$. Moreover, we design a ZKP protocol to prove that the decrypted result is correct with the ciphertext computed by SC using Paillier’s algorithm.

Through the above analysis, it is concluded that using a combination of HE, SC, and ZKP to build the two-party secure operation is a very useful and efficient way to satisfy the security goals of P2P energy trading.

## 5. Results

This section presents numerical results for performance evaluation of the proposed privacy-preserving, P2P negotiation mechanism using different case studies. The case studies were conducted on a computer with an Intel Core i7 processor running at 2.90 GHz and 32 GB RAM. We use Ganache to set up a private Ethereum Homestead blockchain test network. Remote procedure calls via Web3.py/HTTP allow the Python scripts to communicate with the SCs. The Solidity language is used to develop the SCs, which is a special language for SCs on Ethereum.

### 5.1. Simulation Setup

For illustration and discussion, a small distribution network with seven agents is considered as in [33]. The convergence performance, line congestion management, and encryption algorithm performance are shown in Figure 4, Figure 5 and Figure 6, respectively. Then, we investigate the impact of the number of agents on convergence performance, as measured by the number of iterations and computation time, and the results are shown in Figure 7. The results verify that our proposed mechanism is feasible for networks with a large number of agents. For line congestion management, the verification results in networks with 13 nodes are sufficient to prove the feasibility of the proposed mechanism in large networks. Finally, regarding the performance of the encryption algorithm, increasing the number of nodes has little impact on the computational performance since the method is used for the negotiation process between two agents.

There are seven agents in the power network, consisting of four sellers and three buyers. The test system is a 13-node network, as shown in Figure 8. The sellers are located at buses 2, 5, 8, and 10, and the buyers are located at buses 3, 4, and 9. Bus 1 is the reference bus. The connections indicate the physical electrical connections, and the communication network is assumed to have a connected network for the communication of all agents. The parameters of sellers and buyers are listed in Table 1. We set the susceptance of each branch to $b_{1}=b_{2}=\ldots =b_{L}=10s$. All stopping criteria χ are set to $10^{-4}$. The tuning parameters are chosen as follows:(29)$$\delta^{k}=0.1,\;\;\beta^{k}=\frac{0.1}{k^{0.1}},\;\alpha^{k}=\frac{0.1}{k^{0.01}},\;\phi^{k}=10$$
and the stopping criteria are set to
(30)$$\chi^{E}=0.01,\;\chi^{\lambda }=0.01,\;\chi^{\upsilon }=0.01$$

### 5.2. Convergence Performance of the Negotiation Mechanism

In this case study, the maximum line capacity $P_{l}^{max}$ for all lines is set to 10. The convergence process of the algorithm is shown in Figure 4, from which it can be seen that all trading between sellers and buyers converges after about 160 iterations. Although the consensus-based algorithm requires a minimum amount of information to be exchanged, the main drawback is that the number of iterations to converge can be higher than other methods. It can be seen that the sum of the absolute values of the gap of energy quantity and prices decreases with oscillation, while the sum of the absolute values of the gap of line prices remains at zero since no line is congested. The final traded energy quantities and prices are shown in Table 2. It is noticeable that the results of S1 and S3 are the same because their parameters $a_{n}$ and $b_{n}$ are the same. For B1 and B3, the purchase prices are the same, but the quantities of B3 are higher because the demand of B3 is higher (−8 < −7).

### 5.3. Performance of Line Congestion Management

The impact of line capacity limit on power flow is investigated. The maximum line capacity for these lines ranges from 3 to 8 kW. In the test system, the results are shown only for lines with non-zero power flow. The results are shown in Figure 5, and it is confirmed that the power flows in these lines are always within the maximum line capacity, which means that the proposed algorithm can meet the line flow constraints in the P2P power grid. If there is a congested line in the network, agents will avoid trading over the congested lines because they have to pay additional network charges

### 5.4. Performance of Scalability

In the real world, the P2P energy trading mechanism will be used in power networks with a large number of agents, and the number of transactions will be significant. The computation time and the number of iterations are two key factors that measure the scalability of the mechanism. To demonstrate the scalability of our mechanism, we add more agents to each bus. The parameters of the agents are chosen randomly, while the tuning parameters ($\delta^{k},\beta^{k},\mathrm{and}\;\alpha^{k}$) are carefully designed for tolerable performance. The line capacity is chosen large enough to make no congestion happens. Figure 7 shows the effects of the number of agents (between 70 and 420) on the two factors. It can be seen that both the computation time and the number of iterations increase approximately linearly with the number of agents. The performance of computational time is excellent (under 4 s for 420 agents), but more iterations (almost 450) cost. The results show that our proposed mechanism is feasible for networks with a large number of agents.

### 5.5. Encryption Algorithm Computation Performance Analysis

In this section, we analyze the trade-off between privacy and computational cost. In the original decentralized negotiation mechanism, where no homomorphic encryption is applied, the computation time of each agent for each iteration is so small that it is negligible. To ensure privacy, a privacy-preserving mechanism based on homomorphic encryption is proposed to be used at each iteration. Agents need to encrypt their private information and to submit it to SC to perform ciphertext computation. The Paillier homomorphic encryption used in the simulation comes from the phe (Partially Homomorphic Encryption) library in Python. Figure 6a shows the encryption and decryption time of the agents. The encryption time of agent *n* and *m* is close to each other and is about 0.11 s. The decryption time is much lower compared to the encryption time and is about 0.03 s. Figure 6b shows the public/private key and the size of the ciphertext. The size of the ciphertext is slightly larger than 1750, while the public/private keys are much smaller.

### 5.6. Computational Performance under Different Mechanisms

In this section, we investigate the computational performance under four different mechanisms. (1) P2P trading is performed without a privacy-preserving mechanism. (2) P2P trading runs under the Paillier HE mechanism. The agents each encrypt their bid information $\left\{\lambda_{nm},E_{nm}\right\}$ and send it to a program to perform cipher computation. (3) P2P trading runs under the two-party, secure computation mechanism without ZKP. (4) P2P trading runs under the two-party, secure computation mechanism with ZKP. The computation time for each agent in one iteration is displayed in Figure 9. It can be seen that the computation efficiency is very high without any privacy mechanism. The time spent on the second mechanism is higher than for the third because more information needs to be encrypted, which is very time-consuming. The efficiency of the third mechanism is at a medium level and is acceptable. The agents only need to encrypt the aggregated information $\left\{I_{nm},I_{mn}\right\}$, which can greatly reduce the time consumption. Finally, the fourth mechanism is the most ineffective one because the ZKP protocol is very time-consuming and, most of the time, is for computing the inverse element by the expand Euclid algorithm ($M=N^{-1}\;mod\;\phi \left(N\right)$). This problem will be studied in our future work.

### 5.7. Blockchain-Based P2P Energy-Trading Platform

In our simulation, we use Ganache to establish a private Ethereum homestead blockchain named ”Privacy-Preserving P2P Market”, as shown in Figure 10. The first address belongs to MO; the second is LM’s address. The remaining addresses are assigned to each agent. The local computation steps are performed by Python’s codes, and then, the encrypted information is sent via Web3.py/HTTP to SC installed on Ganache.

The agents have two ways to update the energy prices. The first is to run SC $SC_{AG}$ to automatically update the energy prices. The second option is to submit the encrypted information to SC $SC_{AGHE}$ to implement the ciphertext calculation. After updating the energy quantities and prices disseminated over the network, LM updates the power flows and line prices via SC $SC_{LM}$, while MO checks whether the market converges via $SC_{CO}$. Finally, after all trades are balanced, MO stores the transaction results on the blockchain via $SC_{TR}$, which can be checked by anyone on the network.

## 6. Discussion

The most valuable achievement of our proposed mechanism is to provide a privacy-preserving, two-party, secure computation mechanism for the P2P negotiation mechanism between each pair of agents. The agents cannot know each other’s actual bidding information. However, operational efficiency has been sacrificed for privacy protection. A lot of time and computing power are spent on encrypting and decrypting information. In addition, the introduction of SC further extends the time to achieve convergence.

Therefore, our future work will mainly focus on how to increase the computational efficiency under the privacy-friendly mechanism. The first way is to develop a P2P negotiation mechanism that uses a more efficient decentralized optimization algorithm. For example, the consensus ADMM algorithm [31,32], which can guarantee convergence with a smaller number of iterations. The challenge is to combine the consensus ADMM with the HE mechanism. Another way to increase efficiency is to reduce the amount of information to be encrypted or protected. As we analyzed in Section 3.1, in the C + I method, private information is revealed and disclosed only in the collusion attack by all neighboring agents. If we carefully select a part of the exchanged information to be encrypted, the private information can also be protected. We can perform the two-party secure computation with only one neighboring agent, and that is enough to protect private information from attacks. With this strategy, the computation cost can be reduced from $O(N^{2}\Delta T)$ to O(NΔT), where ΔT is the sum of the encryption and decryption time of the two-party secure computation.

## 7. Conclusions

In the P2P energy market, agents must exchange a large amount of information to reach consensus on the final trade. However, this fully decentralized negotiation may lead to the disclosure of private information. In this paper, we propose a privacy-preserving, two-party, secure computation mechanism for P2P energy trading that leverages many technologies. We first design a P2P negotiation mechanism based on the C + I method and the PTDF model. This mechanism can maximize social welfare while satisfying the physical line flow constraints. Then, for this mechanism, we analyze the two collusion attack strategies to obtain private information from a group of malicious neighboring agents. To protect against this kind of attacks, a two-party, secure computation mechanism is proposed for each pair of agents to update the energy prices. The agents first aggregate their bid price and bid quantity and then encrypt the information with the public key generated by the Paillier algorithm. Then, the computation of the ciphertext is automatically performed by SC, and the correctness of the decryption is proved by a ZKP protocol. The simulation results demonstrate the performance of convergence, line congestion management, scalability, computation efficiency, and SC operations.

## Figures and Tables

**Figure 1 sensors-22-09020-f001:**
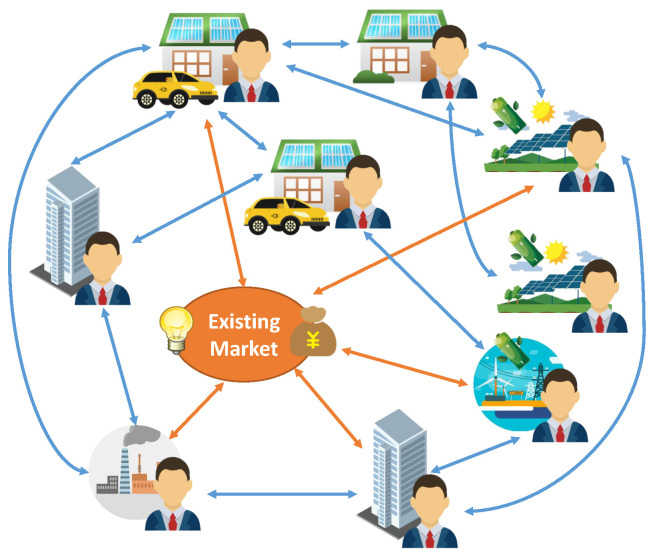
P2P energy-trading market architecture.

**Figure 2 sensors-22-09020-f002:**
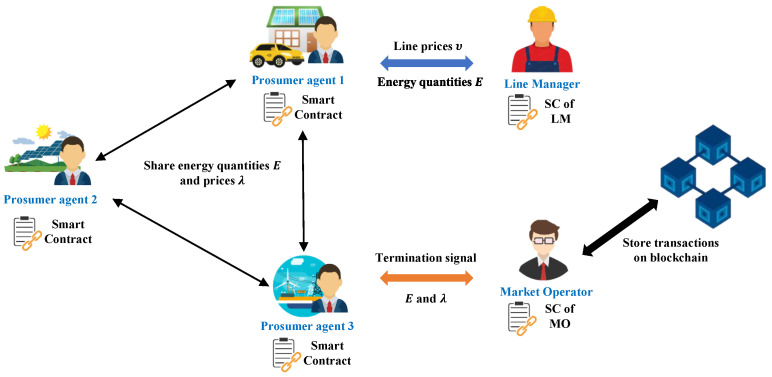
Blockchain-based P2P energy-trading market architecture.

**Figure 3 sensors-22-09020-f003:**
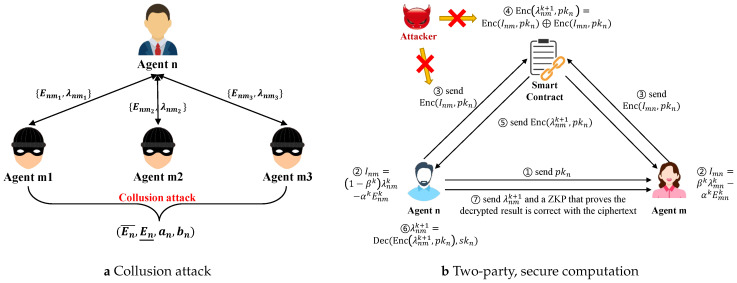
Collusion attack for malicious neighboring agents and two-party, secure computation between two agents.

**Figure 4 sensors-22-09020-f004:**
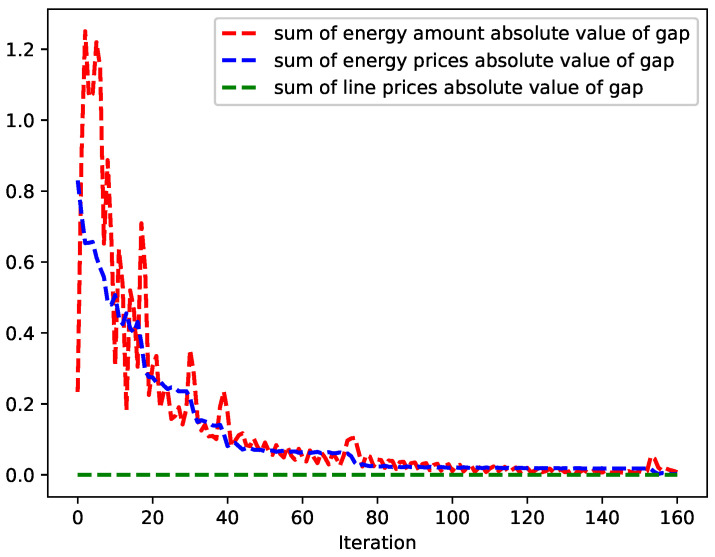
Convergence of the algorithm.

**Figure 5 sensors-22-09020-f005:**
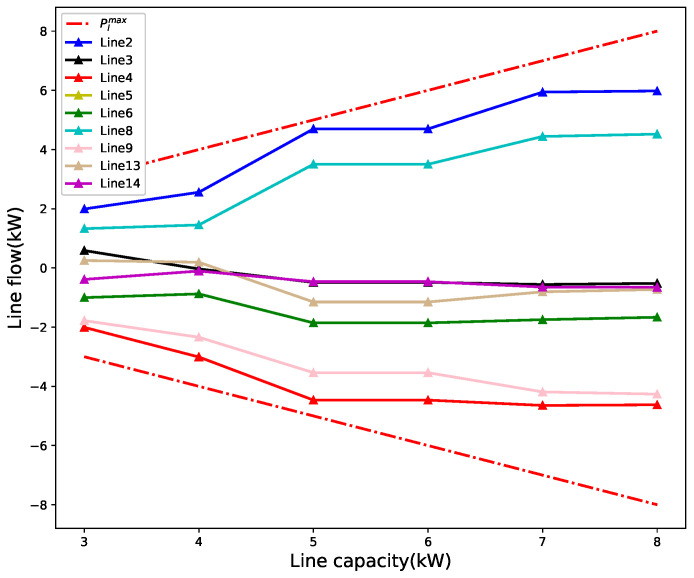
Power flow in different lines under different line capacities.

**Figure 6 sensors-22-09020-f006:**
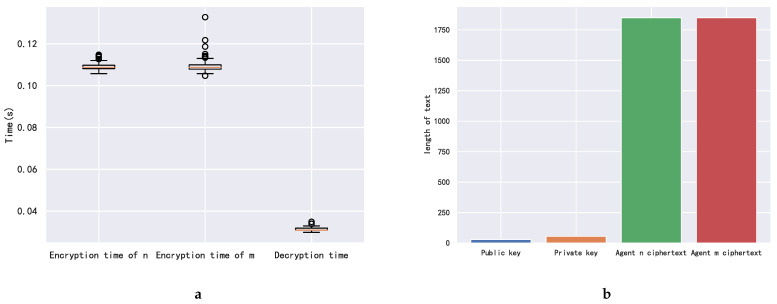
Encryption algorithm computation performance. (**a**) Agent encryption and decryption time. (**b**) The size of the public/private keys and ciphertext.

**Figure 7 sensors-22-09020-f007:**
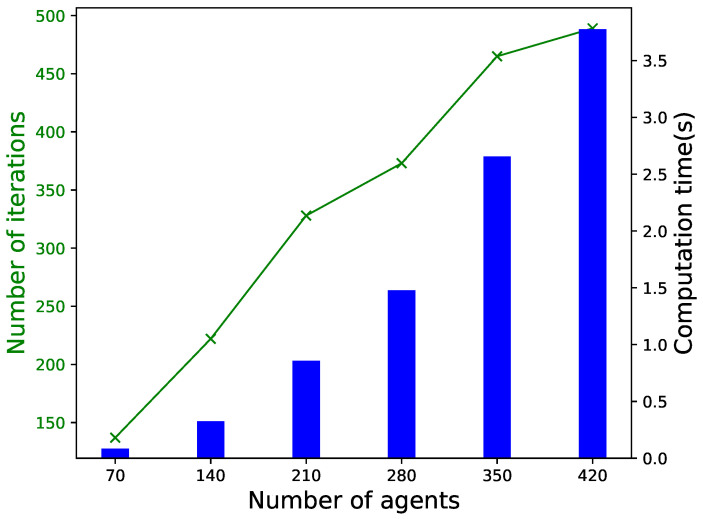
Impact of number of agents on computation time and number of iterations for convergence.

**Figure 8 sensors-22-09020-f008:**
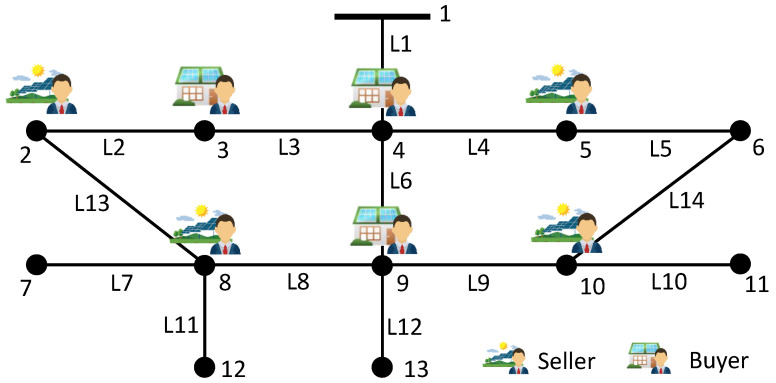
Test system schematic.

**Figure 9 sensors-22-09020-f009:**
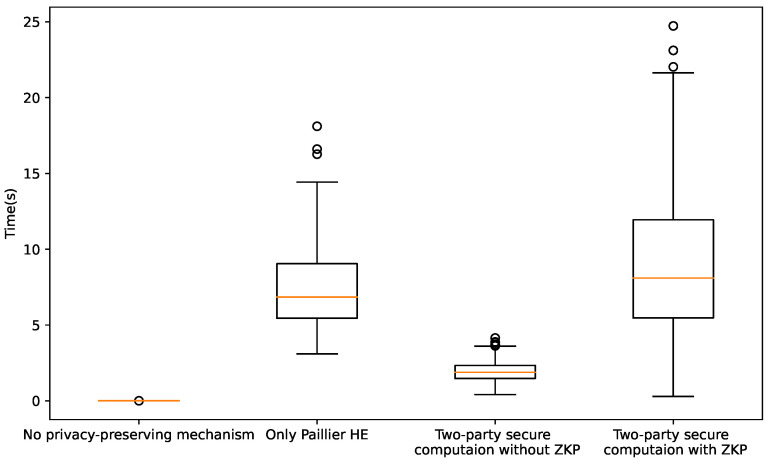
Computational time of different P2P negotiation mechanisms.

**Figure 10 sensors-22-09020-f010:**
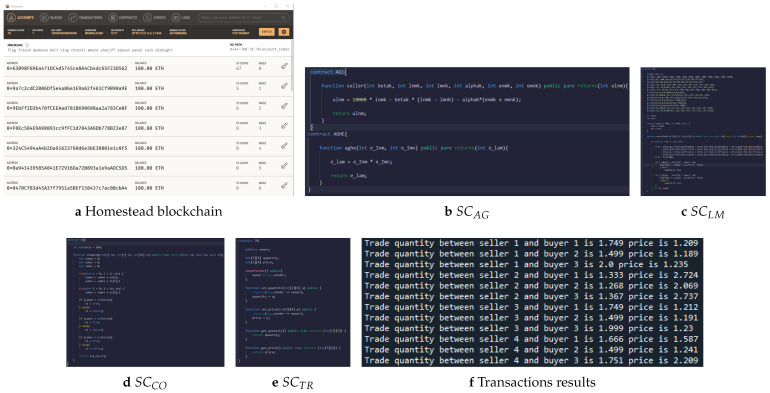
P2P energy-trading blockchain, smart contracts, and transactions results stored on the blockchain.

**Table 1 sensors-22-09020-t001:** Sellers’ and buyers’ parameters of a simple case study.

Agent	Bus	*a_n_* (USD/kW^2^)	*b_n_* (USD/kW)	$\underline{E_{n}}$ (kW)	$\bar{E_{n}}$ (kW)
S1	2	0.04	1	1	7
S2	5	0.046	1	1	4
S3	8	0.04	1	1	6
S4	10	0.05	1	1	5
B1	3	0.05	3	−7	−1
B2	4	0.056	3	−6	−1
B3	9	0.05	3	−8	−1

**Table 2 sensors-22-09020-t002:** Final traded quantities and prices of energy.

	B1	B2	B3
S1	1.75 kW/1.21 USD/kW	1.50 kW/1.19 USD/kW	2.00 kW/1.24 USD/kW
S2	1.33 kW/2.74 USD/kW	1.27 kW/2.72 USD/kW	1.37 kW/2.74 USD/kW
S3	1.75 kW/1.21 USD/kW	1.50 kW/1.19 USD/kW	2.00 kW/1.24 USD/kW
S4	1.67 kW/2.66 USD/kW	1.50 kW/1.24 USD/kW	1.75 kW/2.65 USD/kW

## Data Availability

Not applicable.

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
