# Peer review of "A Privacy-Preserving, Two-Party, Secure Computation Mechanism for Consensus-Based Peer-to-Peer Energy Trading in the Smart Grid"

_sensors, 2022, doi:10.3390/s22229020_

Round 1

Reviewer 1 Report (Previous Reviewer 1)

The questions raised by the reviewer  before have been seriously considered and resolved.  So I suggest to accept the papper in present form.

Author Response

We are grateful to you for your effort reviewing our paper and your positive feedback.

Reviewer 2 Report (Previous Reviewer 2)

Further discussion seems to be necessary on the following issues about comment 2. In section 3, two security goals for the privacy-friendly P2P computation framework are suggested. 

The first goal is to protect the individual privacy information F^k_nm = {E^k_nm, lambda^k_nm}. 

However, the definition fo the first goal is not clear. 

Even using HE, the message from agent m to SC is public so the agent n can decrypt the message to obtain I_mn. It means that the agent n can obtain all of alpha^k, beta^k, and E^k_mn so does lambda^k_mn. If the security goal is just against outside adversaries then usign HE and SC is not required.

The second goal is to guarantee that the third party will obey the energy prices update rules to operate, i.e. making the agent n obey the proper computation rules. 

Note that using BC and SC described in the sec 3.3 guarantees only the computation of ciphertexts, i.e. not for the plaintext-result. The final result lambda^{k+1}_nm is encrypted after the SC computation so that no one except the agent n, who has the secret key, can verify the correctness. 

If you want to construct a two-party secure computation to satisfy this security goal, it requires to apply more complicated methods, like zero-knowledge proof systems. 

Author Response

We are grateful to you for your effort reviewing our paper. Please refer to the attached file for the response to your comments.

Reviewer 3 Report (New Reviewer)

Author Response

We are grateful to you for your effort reviewing our paper. Please refer to the attached file for the reponse to your comments.

Reviewer 4 Report (New Reviewer)

This paper presents a novel P2P negotiation mechanism for energy trading is first proposed based on consensus + innovation method, which only requires a minimum set of information to be exchanged. Furthermore, this paper proposes a two-party secure computation mechanism based on Paillier homomorphic encryption algorithm, blockchain and smart contract, which have emerged as efficient technologies for the implementation of secure and reliable P2P energy trading platform. 

The paper is structured correctly.

The survey presented in the paper is a very interesting and useful contribution to solve the privacy information from attacking by malicious agents, as well as prevent the collusion attack strategy from malicious neighboring agents to steal the privacy information. These contributions were well formulated and the answers to them were correct and clear. I think that this paper is appropriate for publishing. 

Author Response

We are grateful to you for your effort reviewing our paper and your positive feedback.

Round 2

Reviewer 3 Report (New Reviewer)

I appreciate the author's efforts in improving the quality of the manuscript.

Author Response

We are grateful to you for your effort reviewing our paper and your positive feedback.

This manuscript is a resubmission of an earlier submission. The following is a list of the peer review reports and author responses from that submission.

Round 1

Reviewer 1 Report

This work proposed a privacy-preserving mechanism for P2P electricity trading based on Paillier semi-HE method and blockchain technology. However, these drawbacks should be considered.

1.    The authors choose the C+I method to design the market clearing algorithm but did not provide a sufficient reason for why choosing this method.

2.    The computational burden for the information encryption and decryption will be extremely high under the P2P network. How to solve these problems?

3.    In the two-party secure computation mechanism, each agent will make an aggregation operation for individual information. It is better to implement this process also by homomorphic encryption.

4.    The reviewer suggests investigating the computational performance under more agents in the system to demonstrate the scalability.

5.    What is the merit and demerit of the homomorphic encryption method, compared with other privacy-protection approaches, e.g., differential privacy?

6.    Given that there are so many variables and notations in the paper, please give a nomenclature.

The reviewer suggests a thorough revision of the writing.

Reviewer 2 Report

This paper presented a method of energy trading mechanism in P2P environment, especially, the main result seems to be a privacy protecting method using HE and SC. 

There are some comments about the paper 

- a) It seems that the difference from the previous results should be concretely presented. Especially, about for chapter 2 and 3.

- b) It needs proper proof about sec 3.3 two-party secure computation.  

      The author needs to define a concrete security goal for the secure two-party computation. To satisfy the confidentiality suggested in the paper, using combination of HE and SC is useless. 

        - 'I_nm' is reveal to 'user n' anyway. 

        - just sending 'I_nm' to 'user n' after standard public-key encrypting is the simpler method with the same result.

        - also using SC requires verifying the authenticity of the message Enc(I_nm, pk_n)

        -  

- c)